# Operating bedside cardiac ultrasound program in emergency medicine residency: A retrospective observation study from the perspective of performance improvement

Ki Hong Kim[☯], Jae Yun Jung[iD]*[☯], Joong Wan Park[‡], Min Sung Lee[iD][‡], Yong Hee Lee[‡]

Department of Emergency Medicine, Seoul National University Hospital, Seoul, South Korea

☯ These authors contributed equally to this work.
‡ These authors also contributed equally to this work.
* matewoos@gmail.com

**Data Availability Statement:** All relevant data are within the manuscript and its Supporting Information files.

## Abstract

### Background

Point-of-care ultrasound is one of useful diagnostic tools in emergency medicine practice and considerably depends on physician's performance. This study was performed to evaluate performance improvements and favorable attitudes through structured cardiac ultrasound program for emergency medicine residents.

### Methods

Retrospective observational study using the point-of-care ultrasound (PoCUS) database in one tertiary academic-teaching hospital emergency department has been conducted. Cardiac ultrasound education and rotation program has been implemented in emergency medicine residency program. Structured evaluation sheet for cardiac ultrasound and questionnaire toward PoCUS have been developed. An early-phase and a late-phase case were selected randomly for each participant. Two emergency medicine specialists with expertise in PoCUS evaluated saved images independently. We used a paired t-test to compare the performance score of each phase and the results of the questionnaire. Multivariable linear regression analysis was conducted to evaluate the association between the characteristics of participants and performance improvements.

### Results

During the study period, a total of 1,652 bedside cardiac ultrasounds were administered. Forty-six examinations conducted by 23 emergency medicine residents were randomly selected for analysis. The performance score increased from 39.5 to 56.1 according to expert A and 45.3 to 62.9 according to expert B (p-value <0.01 for both). The average questionnaire score, which was analyzed for 17 participants, showed improvement from 18.9 to 20.7 (p-value <0.01). In multivariable linear regression analysis, younger age, higher early-

**Funding:** The authors received no specific funding for this work.

**Competing interests:** The authors have declared that no competing interests exist.

phase score and higher confidence had a negative association with a greater improvement of performance, while the number of examinations had a positive association.

## Conclusions

Bedside cardiac ultrasound performance and attitudes toward PoCUS have been improved through structured residency program.

## Introduction

Point-of-care ultrasound (PoCUS) has been used frequently and variously in emergency clinical practice because it can reduce cost [1] and can be used as an additional diagnostic test [2] that provides important clinical information in a very short time. Procedures have also been proven to benefit from PoCUS, e.g., both guidance and confirmation [3, 4]. Because ultrasound performance depends on the physician's ability, PoCUS training should be a core component in the education of emergency physicians, especially those who are in residency programs [5]. Adding PoCUS training to an education program is considered an important milestone, [6] and many programs have implemented such training, demonstrating feasibility and participant satisfaction [7, 8]. In the United States, PoCUS has been requirement of residency training in emergency medicine [9].

Cardiac ultrasound would be a good performance indicator of PoCUS due to its well established standards for application and evaluation [10]. Furthermore, cardiac ultrasound is known to be useful for diagnosis in emergency departments [11–14]. A recent study by Davood et al. showed that emergency medicine residents can perform bedside cardiac ultrasound in the emergency department after several workshops and that this procedure yields comparable quality to traditional cardiac ultrasound performed by cardiologists [15].

Improvements in practitioners' skills, including in acquiring proper images and interpreting the results, have not been well investigated in previous studies. The evaluation tools are usually limited to objective structured clinical exams (OSCEs) or knowledge tests [16, 17]. In several studies, the diagnostic performance of clinical competency has usually been used to evaluate practitioner performance [7, 18]. To the best of our knowledge, methods to evaluate the performance level of bedside cardiac ultrasound in a real clinical emergency setting have not been developed well.

The purpose of this study is to evaluate the improvement of performance skill and interpretation level of bedside cardiac ultrasound among emergency medicine residents following a PoCUS education-rotation program, by reviewing acquired cardiac ultrasound images. The secondary outcome was a change in attitude and confidence toward PoCUS in emergency practice. We hypothesized that structured program for education and rotation would improves performance and results in favorable attitudes.

## Methods

### Study design and setting

A retrospective observational study was conducted based on the ultrasound database of the Seoul National University Hospital Department of Emergency Medicine. Seoul National University Hospital is a tertiary academic-teaching hospital in the metropolitan city of Seoul, and approximately 70,000 patients visit the emergency department annually. Emergency bedside ultrasound rotation schedule with education program has been in place since 2018, as part of a

residency training program. The goal of this program is to improve the quality of clinical practice. Before implementation, ultrasound was an optional procedure in the emergency department that was conducted when physicians were willing to participate, which was not frequent enough to make ultrasound a standard part of clinical practice. The sample for this study comprised 2nd- to 4th-year residents, who were required to participate in the PoCUS program since it became a formal part of the curriculum and practice protocol in the department.

## Study population

Emergency medicine residents who followed emergency bedside ultrasound rotation duties during the study period, were enrolled. Participants who did not save the whole basic views of ultrasound imaging in the picture archiving communication system or document their interpretations in medical records were excluded from the analysis.

## PoCUS program

The PoCUS program, consisting of an education section and a practice section, was officially designed and implemented in April 2018. All residents had to take a comprehensive ultrasound workshop for basic echocardiography, including basic view, by a cardiologist early in their 2nd years. They then automatically participated in monthly PoCUS education programs if they had been assigned to adult or pediatric emergency departments in Seoul National University Hospital. Residents on other schedules, e.g., in other hospitals or departments, did not participate. The monthly education program included a 2-hour training session and a conference. In the training session, basic and advanced knowledge of PoCUS administration, including for the lung or abdomen, was reviewed, and the participants received hands-on practice administering bedside cardiac ultrasound using an ultrasound simulation machine (US Mentor, Simbionix). Interesting cases were selected by the residents as cases supporting the clinical usability of PoCUS and presented in a conference at the end of the month.

The PoCUS practice session was a requirement of the residency program. All residents had approximately 8–16 hours of ultrasound duties in the emergency department each month. They performed examinations whenever patients presented with chest pain, difficulty breathing, syncope or palpitation. Primary physicians could also request bedside cardiac ultrasounds to residents who are in ultrasound rotation. Residents do not care for patients as primary physicians while on an ultrasound rotation duty. A portable ultrasound machine was used for all procedures (M-turbo, SonoSite) (Vivid Q, GE). Video clips and images were saved and transferred to a picture archiving communication system. Official interpretations of the results were typed in a constructed format. PoCUS rounding was conducted every other day with PoCUS faculty and residents for quality improvement and feedback. All contents were reviewed and supported by the PoCUS faculty, which consisted of 4 emergency medicine specialists and an emergent medical technician. PoCUS faculty need to undergo comprehensive ultrasound education workshops regularly and perform bedside ultrasound in official emergency practice.

## Data source and acquisition

The database system was operated and supervised by the PoCUS faculty using a structured registry that contains demographic and clinical information of patients. It includes date of examination, operator's information and official interpretation.

We retrieved data from the PoCUS database to evaluate performance improvements. Two cases were selected for each resident, one in the first week of the whole period of PoCUS rotation program (early phase) and the other in the last (late phase). Independent research coordinators selected all cases randomly in each period.

## Evaluation for performance

A structured evaluation sheet was developed based on the guidelines of emergency ultrasound standard reporting [19]. The evaluation sheet was designed to evaluate handling the machine, capturing feasible views for interpretation, adjusting depth and gain properly, achieving a complete set of views and interpreting clearly. A detailed introduction to the evaluation sheet is described in Table 1.

Two independent emergency specialists were consulted to evaluate the cases. Two independent emergency specialists were consulted to evaluate the cases. They are PoCUS faculty members who have practical experience with more than 200 cardiac PoCUS and work as instructors for emergency physician and medical school student. Evaluation was based on video clips in a picture archiving communication system (PACS) and developed sheet. The specialists were blind to practitioner identity and phase. Specialists gave specific scores to 6 components of 7 basic views (parasternal long axis, parasternal short axis for 4 levels, 4-chamber and 5 chamber view) and 2 measure indexes (inferior vena cava and ejection fraction) based on the guidelines of the evaluation sheet, and the total score was calculated. One point was given for probe and orientation selection, 2 points were given for adjusting brightness properly, and 3 points were given for imaging well-identified structures of heart, providing a clinically sufficient interpretation and capturing high quality images. The maximum total score was 100 points in each case.

## Survey of confidence and acceptance

A structured questionnaire that consisted of 7 questions regarding confidence in and attitudes about PoCUS use in emergency practice was developed. Each question is answered on a 5-point Likert scale of agreement. Detailed information on the questionnaire is provided in

**Table 1. Evaluation sheet for performance evaluation of cardiac ultrasound in PoCUS program.**

| View | Probe | Utility | Orientation | Anatomy | Interpretation | Image quality |
|---|---|---|---|---|---|---|
| Parasternal Long Axis View | | | | | | |
| Parasternal Short Axis View | | | | | | |
| • AV level | | | | | | |
| • MV level | | | | | | |
| • Papillary muscle level | | | | | | |
| • Apex level | | | | | | |
| Apical 4-Chamber View | | | | | | |
| Apical 5-Chamber View | | | | | | |
| Measure Index | | | | | | |
| • IVC | | | | | | |
| • EF | | | | | | |

PoCUS, point-of-care ultrasound; AV, Aortic valve; MV, Mitral valve; IVC, Inferior vena cava; EF, Ejection fraction

Probe, Appropriate probe was used, score 1; Utility, Total gain adjustment, Time gain adjustment, Focus and depth are well controlled, score 2; Utility, 1 or 2 options have not been adjusted properly, score 1; Utility, 3 or 4 options have not been adjusted properly, score 0; Orientation, Orientation marker was positioned well, score 1; Anatomy, All structures were well identified, score 3; Anatomy, Any core structures was not identified, score 2; Anatomy, Only one structure was identified, score 1; Anatomy, Missed or could not determine specific view, score 0; Interpretation, All important finding has been documented, score 3; Interpretation, Several important findings have been documented, score 2; Interpretation, Some interpretation was inappropriate, score 1; Interpretation, No specific interpretation for view, score 0; Image quality, Seems no need for improvement, score 3; Image quality, Fine but need some improvement, score 2; Image quality, Can recognize specific view, but limited to interpret, score 1; Image quality, Cannot recognize specific view, score 0

**Table 2. Questionnaire for survey of PoCUS program.**

| | |
|---|---|
| 1. Can you use ultrasound in emergency department which you are working? | |
| (including select probe, adjust gain and depth, saving image) | |
| Response set | Strongly Disagree—Disagree—Neutral—Agree—Strongly Agree |
| 2. How many patients are you applying ultrasound for one duty in daytime? | |
| Response set | less than two—two—three—four—more than four |
| 3. How many patients are you applying ultrasound for one duty in nighttime? | |
| Response set | less than two—two—three—four—more than four |
| 4. How much proportions of "Core Basic View" can you acquire properly in adult cardiac ultrasound? | |
| (Parasternal long axis, Parasternal short axis, 4-chamber view, 5-chamber view, IVC) | |
| Response set | 0~20% - 20~40% - 40~60% - 60~80% - 80~100% |
| 5. How much proportions of "Core Basic View" can you acquire properly in adult abdominal ultrasound? | |
| (Liver, GB, Spleen, Kidney, Bladder, Abdominal Aorta) | |
| Response set | 0~20% - 20~40% - 40~60% - 60~80% - 80~100% |
| 6. How much proportions of "Core Basic View" can you acquire properly in pediatric abdominal ultrasound? | |
| (Liver, GB, Spleen, Kidney, Bladder, Abdominal Aorta) | |
| Response set | 0~20% - 20~40% - 40~60% - 60~80% - 80~100% |
| 7. This question is about clinical utility of point-of-care ultrasound in emergency setting. How many ultrasounds will you perform in one duty after residency program? | |
| Response set | less than two—two—three—four—more than four |

* Response set, Likert 5-scale (1-2-3-4-5)

PoCUS, point-of-care ultrasound; IVC, inferior vena cava; GB, gallbladder

Table 2. All participants completed the questionnaire at the beginning and end of the PoCUS program. This questionnaire was used to judge quality improvements and applicability to residency programs.

## Statistical analyses

A descriptive analysis was conducted by calculating the mean and standard deviation for the demographics and characteristics of the participants. Total early- and late-phase scores were compared by paired t-test for each participant, and the intraclass correlation coefficient was calculated. Sensitivity analysis for each basic core view and questionnaire analysis were conducted using the same methods.

An additional multivariable linear regression analysis was conducted to evaluate the association between the demographics and characteristics of the participants and score improvements. A stepwise method was used to develop the optimized model. Score improvement was defined as the mean difference between each phase. We selected predictors such as year of graduation (YOG), age, time elapsed between assessments, early-phase score, number of PoCUS applications, duration of program participation and answers to the questionnaires about confidence in and acceptance of bedside cardiac ultrasound. All statistical analyses were conducted in R Studio 4.3.4.

## Ethics approval

Ethics approval and consent to participate: The study complies with the Declaration of Helsinki, and its protocol was approved by the Seoul National University Hospital Institutional Review Board with a waiver of informed consent (IRB No. 1911-076-1078).

## Results

### Characteristics of study subjects

From April 2018 to February 2019, a total of 28 emergency medicine residents participated in the PoCUS program. 46 ultrasound data from 23 participants were analyzed in the performance evaluation, since 5 participants were excluded who did not save the whole basic ultrasound views or interpretations. In the survey analysis, 6 participants refused to fill out questionnaire. 22 residents conducted 35 pairs (35 early phase and 35 late phase) of questionnaires, since 13 residents participated POCUS each year during the study period. The demographics and characteristics of the study participants are described in Table 3. The mean period between the early and late phases, which is almost same as the gap between first and last rotation, was 20 weeks. The mean participation in the PoCUS program was 3 months, which was not the same for each individual due to rotation schedule including dispatching to other hospitals. They administered PoCUS nearly 40 times on average, and there was one exceptional practitioner who conducted 215 cardiac ultrasounds in 5 months.

### Performance and survey evaluation

Table 4 and Fig 1 shows the results of the performance evaluations conducted by two emergency specialists using the structured sheet and questionnaire evaluations. A total of 46 cases from 23 participants were analyzed, and the mean score changed significantly from 39.5 to 56.1 according to expert A and 45.3 to 62.9 according to expert B. The interclass correlation coefficients were 0.85 and 0.84 for the early and late phases, respectively. The total average questionnaire score improved from 18.9 to 20.7. The most improved question was about confidence in acquiring the core basic view using adult cardiac ultrasound (Q4 in Table 2), which changed from 3.14 to 3.83.

**Table 3. Characteristics of study population and bedside cardiac ultrasound.**

| | | N | % |
|---|---|---|---|
| Total | | 23 | |
| YOG | | | |
| | 1 | 1 | 4.3 |
| | 2 | 9 | 39.1 |
| | 3 | 8 | 34.8 |
| | 4 | 5 | 21.7 |
| | | Mean | SD |
| Age | | 30.96 | 3.2 |
| Period difference, week | | 20.87 | 10.83 |
| Participated period, month | | 3.04 | 1.15 |
| Score of Early phase | | 42.39 | 19.58 |
| Score of Late Phase | | 59.52 | 19.54 |
| Number of Cardiac Ultrasound Examinations | | 38.78 | 44.38 |
| Prior Answers for Question about Confidence | | 2.76 | 0.97 |
| Prior Answers for Question about Acceptance | | 2.88 | 1.32 |
| Post Answers for Question about Confidence | | 3.88 | 0.7 |
| Post Answers for Question about Acceptance | | 3.24 | 1.39 |

YOG, Year of graduate

**Table 4. Performance evaluation of cardiac ultrasound and acceptance for PoCUS program.**

|  | Expert A | Expert B | ICC, 95% CI | Questionnaire |
|---|---|---|---|---|
| Early phase, mean (SD) | 39.5 (21.2) | 45.3 (19.2) | 0.85 (0.61–0.94) | 18.9 (4.65) |
| Late phase, mean (SD) | 56.1 (21.7) | 62.9 (18.5) | 0.84 (0.52–0.94) | 20.7 (4.51) |
| p-value of Shapiro-Wilk normality test | 0.56 | 0.84 |  |  |
| p-value for paired t-test | <0.01 | <0.01 |  | <0.01 |

PoCUS, point-of-care ultrasound; ICC, Interclass correlation coefficiency; CI, Confidence Interval; SD, Standard deviation

## Sensitivity analysis for detailed components

We conducted additional sensitivity analysis for each component of the bedside cardiac ultrasound core basic view. There was significant improvement in performance based on each expert's evaluation of the parasternal long axis; parasternal short axis, aortic valve; parasternal short axis, mitral valve; parasternal short axis, papillary muscle; and 4-chamber view. The interclass correlation coefficient was mostly between 0.65 and 0.8, except for the inferior vena cava view (0.43 and 0.23) (Table 5).

## Regression model for improvement of performance

In multivariable linear regression analysis, we found that age, early-phase score and prior high confidence had a negative association with performance improvement; the beta coefficients

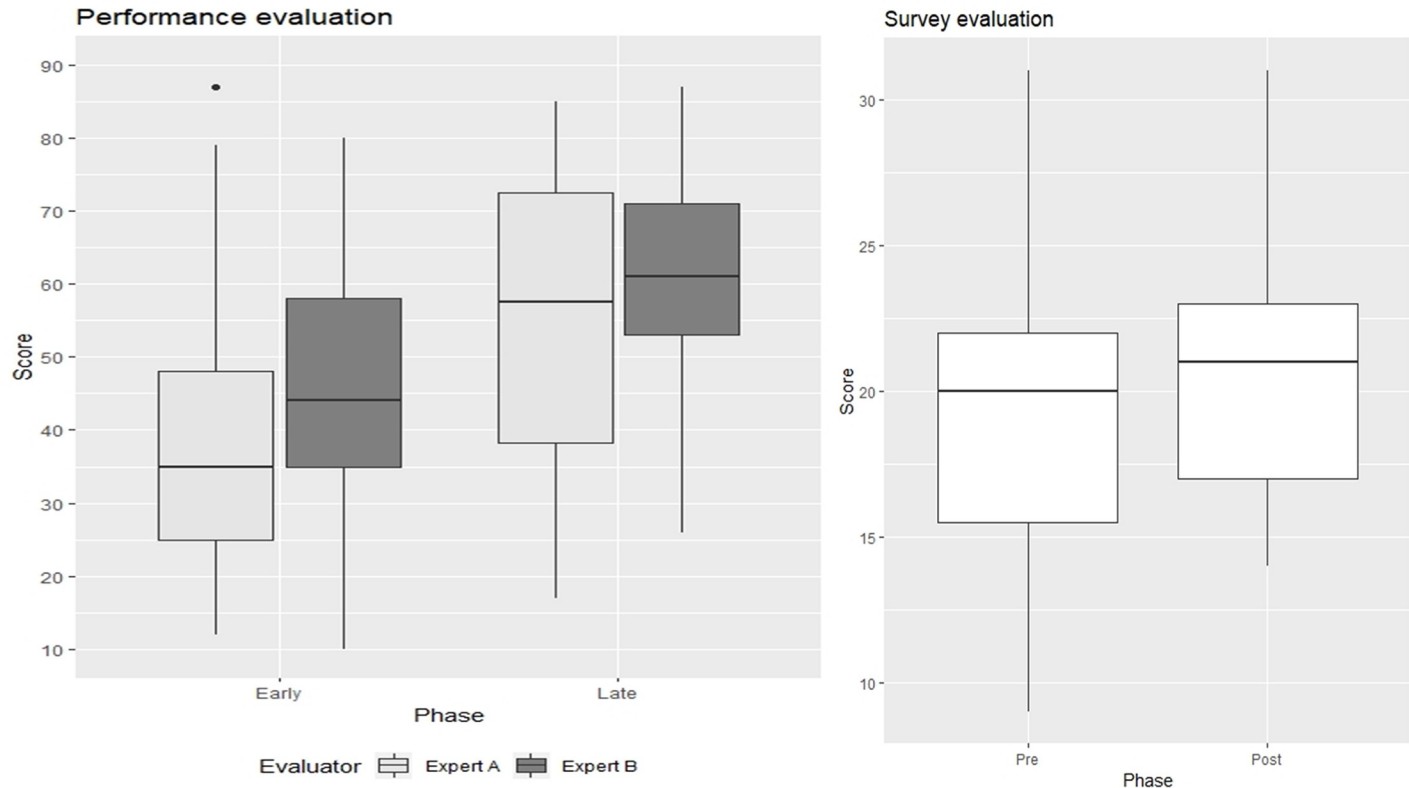

**Fig 1. Evaluation of performance improvement in bedside cardiac ultrasound and survey about confidence and acceptance toward PoCUS.** PoCUS, point-of-care ultrasound.

**Table 5. Sensitivity analysis for cardiac ultrasound performance evaluation.**

| View | Phase | Expert A | Expert B | ICC, 95% CI |
|---|---|---|---|---|
| Parasternal Long Axis | Early phase, mean (SD) | 7.96 (2.67) | 8.26 (2.70) | 0.84 (0.66–0.93) |
| | Late phase, mean (SD) | 10.0 (2.06) | 10.3 (1.75) | 0.65 (0.33–0.83) |
| | p-value for paired t-test | <0.01 | <0.01 | |
| Parasternal Short Axis, AV level | Early phase, mean (SD) | 4.65 (4.48) | 4.91 (3.69) | 0.86 (0.69–0.94) |
| | Late phase, mean (SD) | 7.30 (4.83) | 7.39 (3.74) | 0.65 (0.33–0.84) |
| | p-value for paired t-test | <0.01 | <0.01 | |
| Parasternal Short Axis, MV level | Early phase, mean (SD) | 6.09 (3.87) | 7.78 (3.23) | 0.61 (0.23–0.82) |
| | Late phase, mean (SD) | 8.52 (4.18) | 9.83 (3.52) | 0.77 (0.48–0.90) |
| | p-value for paired t-test | 0.03 | 0.04 | |
| Parasternal Short Axis, Papillary muscle level | Early phase, mean (SD) | 5.70 (4.20) | 6.87 (4.26) | 0.89 (0.64–0.96) |
| | Late phase, mean (SD) | 8.04 (4.81) | 9.65 (3.65) | 0.73 (0.41–0.88) |
| | p-value for paired t-test | 0.04 | <0.01 | |
| Parasternal Short Axis, Apex level | Early phase, mean (SD) | 3.39 (4.19) | 4.43 (4.59) | 0.86 (0.67–0.94) |
| | Late phase, mean (SD) | 5.17 (5.39) | 6.09 (5.73) | 0.77 (0.54–0.90) |
| | p-value for paired t-test | 0.18 | 0.25 | |
| 4-Chamber View | Early phase, mean (SD) | 3.87 (4.30) | 4.83 (3.89) | 0.86 (0.67–0.94) |
| | Late phase, mean (SD) | 7.26 (4.39) | 8.13 (4.35) | 0.77 (0.54–0.90) |
| | p-value for paired t-test | <0.01 | <0.01 | |
| 5-Chamber View | Early phase, mean (SD) | 2.96 (3.67) | 3.39 (4.35) | 0.86 (0.67–0.94) |
| | Late phase, mean (SD) | 3.74 (4.66) | 4.96 (5.23) | 0.77 (0.54–0.90) |
| | p-value for paired t-test | 0.51 | 0.28 | |
| Inferior Vena Cava View | Early phase, mean (SD) | 3.26 (2.56) | 3.78 (2.32) | 0.43 (0.04–0.71) |
| | Late phase, mean (SD) | 3.91 (2.48) | 4.96 (1.64) | 0.23 (0–0.56) |
| | p-value for paired t-test | 0.27 | 0.84 | |

ICC, Interclass correlation coefficiency; CI, Confidence Interval; SD, Standard deviation; AV, Aortic valve; MV, Mitral valve

were -2.6 (-4.8 to -0.4) for age, -0.4 (-0.7 to -0.1) for early-phase score, and -11.0 (-18.6 to -3.4) for prior answers to questions about confidence. Otherwise, the number of cardiac ultrasound examinations had a positive association with improved performance, with a beta coefficient of 0.4 (0.2 to 0.5) (Table 6).

## Discussion

We found statistically significant improvement of cardiac ultrasound performance by emergency medicine residents following structured education and rotation programs. Furthermore,

**Table 6. Multivariable linear regression analysis for performance improvement in bedside cardiac ultrasound.**

| Variables | ß | 95% CI | p-value |
|---|---|---|---|
| YOG | 9.4 | -0.8 to 19.6 | 0.066 |
| Age | -2.6 | -4.8 to -0.4 | 0.027 |
| period difference, week | 0.5 | -0.1 to 1.1 | 0.1 |
| Score of Early phase | -0.4 | -0.7 to -0.1 | 0.014 |
| Number of Cardiac Ultrasound Examinations | 0.4 | 0.2 to 0.5 | <0.001 |
| Prior Answers for Question about Confidence | -11.0 | -18.6 to -3.4 | 0.009 |

CI, Confidence Interval; YOG, Year of graduate

attitudes about and confidence in emergency PoCUS application improved. Additional analysis showed that younger age, lower previous performance level and confidence were associated with marked improvement.

Our study used a paired t-test, which has been successfully used to evaluate various programs for resident education, to evaluate the effect of the program in each participant [20, 21]. Recent research on focused cardiac ultrasound in surgical intensive care units used similar statistical methods and standardized scoresheets [22]. Our scoresheet included similar categories to those of previous studies but had more detailed components and evaluation instructions, which were designed by the PoCUS faculty. The correlation between the two emergency specialists revealed good agreement on total score and most of the sensitivity analysis [23].

Multivariable linear regression analysis was conducted to evaluate the association between the characteristics of participants and potential improvement. Variables could be retrieved only from the clinical database since it was a retrospective observation study, and the results were reasonable and explainable. We may consider focusing on residents with low performance levels and confidence, which would be more effective.

Previous studies have been predominantly based on education programs, and evaluations have not been blinded to both raters and participants. Our evaluation was conducted based on a clinical practice database in an emergency department. We assume that all ultrasound procedures were performed independently for clinical purposes without additional consideration, which would guarantee more objective outcomes than evaluating on via the education program.

There has been little evidence of an effective curriculum and evaluation method for ultrasound training. In focused abdominal sonography for trauma (FAST), previous studies usually focused on requirements for certification or competency [24–26]. The results of this research demonstrated an effective way to increase the performance and favorable attitudes of cardiac PoCUS in residents. Anstey et al. demonstrated an effective PoCUS curriculum for internal medicine residents that produced long-term gains in knowledge and high confidence achievements [27]. Emergency medicine residents struggle in overcrowded emergency departments to manage various patients with severe needs, which results in a stressed and dissatisfied training period [28, 29]. It often leads residents to overload themselves to complete the mandatory cardiac PoCUS training sessions off duty. On the other hand, procedural training without real clinical practice can blur the effect of education programs. Furthermore, a recent study showed that doctors recognize their need for sufficient practice with certain procedures to build competency [30]. We can infer that less education and more practice would help improve both performance and attitude on cardiac PoCUS.

Our research focused on performance improvement of cardiac PoCUS with real practice, but the clinical benefit has not been well studied. We are planning an additional study to evaluate the association between the implementation of cardiac PoCUS protocol for specific population and improvement of clinical outcomes. Additionally, after operating the PoCUS program for a few more years, other components of the PoCUS program, including pediatric abdomen sonography and lung ultrasound, are thought to be analyzed with sufficient case volume.

Our study has several limitations. First, we screened all residents for the study population and included 23 participants based on compatibility for analysis rather than the number of examinations they performed. A previous study demonstrated a cutoff for improvement in ultrasound administration [31]. In our study, residents conducted cardiac ultrasound when needed to evaluate patients, but they did not log all examinations in the database, especially in urgent situations. Accordingly, the number of cases with usable data varied from 10 to over 200 for each participant but confining the study population based on the number of examinations completed may be inappropriate for study purpose of evaluating all participants in the

program. Also, even assuming that the data would have been saved with well-performed ultrasound, the missing ultrasound data could lead to bias, which is one of the major limitations of the study. Second, since the yearly schedule is different for each participant, including the timing of the adult emergency department rotation, the number of program openings per educational period and the number of PoCUS rotations available would differ. The participation period varied from 1 month to 5 months. We assume that if the residents recognize the clinical importance of PoCUS, they will practice it in the course of their regular duties even if they do not need to. Next, scoring system used in this study has not been validated for various environment. Though it was developed based on the standard guidelines, further research for validation is needed. Fourth, evaluation of performance was conducted by emergency medicine specialists. In United States, certification for PoCUS competency in emergency practice has been developed for various institutes. We consider certification of assessor as appropriate with more than 200 cardiac PoCUS experienced, and qualification for PoCUS instructor in Seoul National University Hospital Department of Emergency Medicine. Detailed components to prove the practitioner's credentials may not match perfectly due to lack of official PoCUS certificate programs in South Korea, which can be another limitation. Fifth, this study was limited to one tertiary academic hospital emergency department, and thus cannot be generalized directly to other environments. We are trying to construct a multicenter PoCUS program, which may overcome this limitation. Next, concrete description or analysis for diastolic function and valve function was not conducted. Though some description for them was in interpretation, we considered competency of participants is insufficient, which including diastolic function and valve function seems inappropriate. Last, this is a retrospective observational study, which has limitations in proving the effect of experience.

## Conclusion

We found performance improvements in bedside cardiac ultrasound administration and favorable attitudes toward PoCUS in emergency medicine residents who were enrolled in a structured education and rotation program.

## Supporting information

**S1 Dataset.**
(XLSX)

## Author Contributions

**Conceptualization:** Ki Hong Kim, Jae Yun Jung.

**Data curation:** Ki Hong Kim.

**Formal analysis:** Ki Hong Kim, Joong Wan Park.

**Investigation:** Ki Hong Kim, Jae Yun Jung, Min Sung Lee, Yong Hee Lee.

**Methodology:** Ki Hong Kim, Jae Yun Jung, Joong Wan Park.

**Resources:** Min Sung Lee, Yong Hee Lee.

**Software:** Ki Hong Kim.

**Supervision:** Jae Yun Jung.

**Validation:** Ki Hong Kim, Joong Wan Park, Min Sung Lee, Yong Hee Lee.

**Visualization:** Ki Hong Kim, Yong Hee Lee.

**Writing – original draft:** Ki Hong Kim.

**Writing – review & editing:** Jae Yun Jung, Min Sung Lee.

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
