## [Decision Letter · Decision Letter 0]

8 Jan 2021

PONE-D-20-30438

Operating bedside cardiac ultrasound program in emergency medicine residency: a retrospective observation study from the perspective of performance improvement

PLOS ONE

Dear Dr. Jung,

Thank you for submitting your manuscript to PLOS ONE. After careful consideration, we feel that it has merit but does not fully meet PLOS ONE’s publication criteria as it currently stands. Therefore, we invite you to submit a revised version of the manuscript that addresses the points raised during the review process.

This is an interesting and generally well-done paper in an area with little formal research. As such, I believe it will make an important contribution to the medical literature, provided the recommended changes can be made.

In addition to the reviewer comments, can you please state whether the scores were normally distributed and thus appropriate for analysis with a t-test. i also agree that lack of cardiologist involvement is not a limitation and should be removed.

We look forward to receiving your revised manuscript.

Kind regards,

Robert Ehrman, MD, MS

Academic Editor

PLOS ONE

Reviewers' comments:

Reviewer's Responses to Questions

**Comments to the Author**

1. Is the manuscript technically sound, and do the data support the conclusions?

Reviewer #1: Yes

Reviewer #2: Yes

Reviewer #3: Yes

2. Has the statistical analysis been performed appropriately and rigorously? 

Reviewer #1: Yes

Reviewer #2: Yes

Reviewer #3: Yes

3. Have the authors made all data underlying the findings in their manuscript fully available?

Reviewer #1: Yes

Reviewer #2: Yes

Reviewer #3: Yes

4. Is the manuscript presented in an intelligible fashion and written in standard English?

Reviewer #1: Yes

Reviewer #2: Yes

Reviewer #3: Yes

5. Review Comments to the Author

Reviewer #1: Thanks for the opportunity to review this.

As you say, it shows objective improvement in competence and subjective confidence following a structured POCUS training programme.

I have a couple of queries

- why does the mean time in program (3 months) and the mean period between early and late phases (20 weeks) not match? You state early and late phase are at the beginning and end of the programme?

- I would value some more detail / clarity about the qualifications or experience of the faculty / assessors. it is not clear how experienced or suitable they are at assessing the quality of images (I am sure they are perfectly qualified but to allow reproducibility we need more details of their qualifications)

Reviewer #2: The article is an interesting analysis on the Operating bedside cardiac ultrasound program in emergency medicine residency. The article is clear and the main results well represented. To make the article more appealing for readers would be of interest to know also attitude in the use of color doppler and pulse wave doppler for assessment of diastolic function and have a dedicated issue on assessment of valve disease.

Reviewer #3: Thank you for the opportunity to review this manuscript on a retrospective analysis of the impact of a dedicated POCUS on attitudes and output of a POCUS program in an EM residency. Overall i found the data contained in the manuscript to be informative and very useful for readers who play a role in an EM based POCUS program. Strengths of the manuscript include:

-clearly stated statement of objective

-an detailed enough description of what was done that a similar researcher could reproduce this study in their own EM program

-utilizing independent research coordinators to randomly select the cases from the two periods that were evaluated by the study team

-detailed explanation of the items that were evaluated in each of the cardiac ultrasound views

-employing a sensitivity analysis

THe following items are things that i believe should be addressed to improve the manuscript:

-at the end of the 1st paragraph in the Introduction you should state that the ACGME RRC EM requires/mandates training in POCUS and cite their documents. it would be a more powerful and accurate statement to highlight the importance of POCUS for the general readership

-although the objective statement at the conclusion of the Intro is explicitly stated, i do take issue with the use of the term "perception". It is not clearly defined anywhere in the manuscript and b/c you are including it in your objective statement i think you should either revise the verbiage of the objective statement, or include in the methods section how you intend to measure "perception"

-the use of the term "screening" in the Methods section seems misplaced. It almost seems like you simply excluded residents who didn't follow the parameters of the POCUS rotation, rather than screening. I think you should eliminate the use of the word screening here and simply describe precisely how you went about selecting the participants

-your scoring system, although appearing to be quite robust, is certainly novel. Did you do anything to internally validate this measure?

-going back to the "screening" issue, by my interpretation of the manuscript there were 33 eligible participants amongst the 2nd-4th year residents, and ultimately 22 of them were included b/c 6 refused to complete the questionnaire and 5 were excluded b/c they didn't perform or document appropriately their POCUS exams. If this is indeed true, i think a flow diagram outlining this would be a much better way to describe this then the way it is currently described in the manuscript using the "screening" terminology

- on lines 229-230 of the Results section you state that younger age had a negative association with performance improvement, however on lines 243-244 of the Discussion section you state that younger age had an association with marked improvement. Please resolve this discrepancy.

-since this study was essentially only looking at cardiac POCUS, i would recommend rewording much of the paragraph on lines 263-276

-lines 277-283 seem to be discussing practice of POCUS that occurs "on shift" in an ED, although this is not explicitly stated, i would rephrase to include this term if this is indeed what you are discussing here

-Line 300 referring to the lack of a cardiologist performed evaluation is not a Limitation. Cardiologists are not content experts in cardiac POCUS and need not be the ones performing evaluations of POCUS exams in the ED

6. PLOS authors have the option to publish the peer review history of their article (what does this mean?). If published, this will include your full peer review and any attached files.

Reviewer #1: No

Reviewer #2: No

Reviewer #3: No

---

## [Author Response · Author response to Decision Letter 0]

14 Jan 2021

We sincerely appreciate your comment. I would like to notify you that scores found to be normally distributed by shaprio-wilks test and thus considered to be appropriate for analysis with a t-test.

---

## [Decision Letter · Decision Letter 1]

17 Feb 2021

PONE-D-20-30438R1

Operating bedside cardiac ultrasound program in emergency medicine residency: a retrospective observation study from the perspective of performance improvement

PLOS ONE

Dear Dr. Jung,

Thank you for submitting your manuscript to PLOS ONE. After careful consideration, we feel that it has merit but does not fully meet PLOS ONE’s publication criteria as it currently stands. Therefore, we invite you to submit a revised version of the manuscript that addresses the points raised during the review process.

Thank you for your revised submission, it is overall markedly improved. However, there are still a few areas that require your attention. Please review and explicitly address the concerns raised by reviewer #3. I agree that proper understanding of how these issues were handled is critical to the interpretation and application of the results of your study.

We look forward to receiving your revised manuscript.

Kind regards,

Robert Ehrman, MD, MS

Academic Editor

PLOS ONE

Reviewers' comments:

Reviewer's Responses to Questions

**Comments to the Author**

1. If the authors have adequately addressed your comments raised in a previous round of review and you feel that this manuscript is now acceptable for publication, you may indicate that here to bypass the “Comments to the Author” section, enter your conflict of interest statement in the “Confidential to Editor” section, and submit your "Accept" recommendation.

Reviewer #2: All comments have been addressed

Reviewer #3: (No Response)

2. Is the manuscript technically sound, and do the data support the conclusions?

Reviewer #2: Yes

Reviewer #3: Yes

3. Has the statistical analysis been performed appropriately and rigorously? 

Reviewer #2: Yes

Reviewer #3: Yes

4. Have the authors made all data underlying the findings in their manuscript fully available?

Reviewer #2: Yes

Reviewer #3: Yes

5. Is the manuscript presented in an intelligible fashion and written in standard English?

Reviewer #2: Yes

Reviewer #3: Yes

6. Review Comments to the Author

Reviewer #2: I'm satisfied with revision and aswers provided by authors to the main comments...............................................................................................................................................................................................................................................................

Reviewer #3: Thank you for taking the time to revise your manuscript. I do however have one overarching issue that remains in spite of the revisions, and that is how you handled incomplete data sets. Two specific areas of the revised manuscript In the Methods section, lines 91-93. How did you treat the scenario where a resident saved some of the images of the exams they did, but not all of them? Were they excluded, or did you only exclude residents who did not save any of their images? Selection bias is possible (likely) given the manner in which most residents save some, but not all of their images from exams they do on ultrasound rotations.

The other location in the manuscript where this comes up is in the Results section, lines 194-198. 35 pairs of questionnaires from 22 inhabitants means there was a lot of participants who only completed 1 survey…and also, the 22 participants is different from the 28 participants identified in the first sentence. What happened to the 6 residents w/o surveys? The next sentence says 5 were excluded so that still leaves you at 23 and not 22. The old manuscript says 6 participants refused to complete the questionnaire, why was this info excluded from the revision?

Without more detail on how incomplete data was handled, it is not possible to reproduce this study, and this is why i cannot recommend it for publication in its presents state.

7. PLOS authors have the option to publish the peer review history of their article (what does this mean?). If published, this will include your full peer review and any attached files.

Reviewer #2: No

Reviewer #3: No

---

## [Author Response · Author response to Decision Letter 1]

18 Feb 2021

Reviewer #3: 

Two specific areas of the revised manuscript In the Methods section, lines 91-93. How did you treat the scenario where a resident saved some of the images of the exams they did, but not all of them? Were they excluded, or did you only exclude residents who did not save any of their images? Selection bias is possible (likely) given the manner in which most residents save some, but not all of their images from exams they do on ultrasound rotations.

(ANSWER) Thank you for the review. Participants were excluded who did not saved the whole basic view of ultrasound. The entire set of basic view were introduced and educated to participants at the beginning of POCUS program. We considered that evaluating the best performance at each point is thought to be proper in comparing between phases. We clarified this at the manuscript more clearly. 

(REVISION: Methods) Participants who did not save the whole basic views of ultrasound imaging in the picture archiving communication system or document their interpretations in medical records were excluded from the analysis.

(REVISION: Methods) All residents had to take a comprehensive ultrasound workshop for basic echocardiography, including basic views, by a cardiologist early in their 2nd years.

The other location in the manuscript where this comes up is in the Results section, lines 194-198. 35 pairs of questionnaires from 22 inhabitants means there was a lot of participants who only completed 1 survey…and also, the 22 participants is different from the 28 participants identified in the first sentence. What happened to the 6 residents w/o surveys? The next sentence says 5 were excluded so that still leaves you at 23 and not 22. The old manuscript says 6 participants refused to complete the questionnaire, why was this info excluded from the revision?

(ANSWER) Thank you for your valuable comments. First, 6 residents denied or neglect conducting surveys, as you indicated. 5 participants were excluded from performance analysis, not survey analysis. Total achieved questionnaires were 35 pairs, 1 pair for Early-phase and Late-phase. Which means there were 70 questionnaires for 22participants. There were 13 residents, who participated yearly POCUS program in succession (as described in manuscript, 2nd ~ 4th year of residents participate the program), conducted questionnaire in each year. We clarified this in the manuscript more clearly.

(REVISION: Results) From April 2018 to February 2019, a total of 28 emergency medicine residents participated in the PoCUS program. 46 ultrasound data from 23 participants were analyzed in performance evaluation, since 5 participants were excluded who did not save the whole basic ultrasound views or interpretations. In survey analysis, 6 participants refused to fill out questionnaire and 22 residents conducted 35 pairs (35 early-phase and 35 late phase) of questionnaires, since 13 residents participated yearly POCUS program in succession for study period.

---

## [Editor Report · Decision Letter 2]

23 Feb 2021

PONE-D-20-30438R2

Operating bedside cardiac ultrasound program in emergency medicine residency: a retrospective observation study from the perspective of performance improvement

PLOS ONE

Dear Dr. Jung,

Thank you for submitting your manuscript to PLOS ONE. After careful consideration, we feel that it has merit but does not fully meet PLOS ONE’s publication criteria as it currently stands. Therefore, we invite you to submit a revised version of the manuscript that addresses the points raised during the review process.

Thank you for clarifying the issue regarding missing US exams. However, beyond just simply stating that it exists, I believe that it is important to add some interpretation of the ramifications of the missing data. Does it introduce bias? How does it impact results, etc?

We look forward to receiving your revised manuscript.

Kind regards,

Robert Ehrman, MD, MS

Academic Editor

PLOS ONE
---

## [Author Response · Author response to Decision Letter 2]

23 Feb 2021

Thank you for clarifying the issue regarding missing US exams. However, beyond just simply stating that it exists, I believe that it is important to add some interpretation of the ramifications of the missing data. Does it introduce bias? How does it impact results, etc?

(ANSWER) Thank you for the review. As your valuable comments, missing ultrasound data is one of limitation of this research. We would like to describe the assumption and potential bias of that in the discussion section as below.

(REVISION: Discussion) Also, even assuming that the data would have been saved with well-performed ultrasound, the missing ultrasound data could lead to bias, which is one of the major limitations of the study.

---

## [Editor Report · Decision Letter 3]

4 Mar 2021

Operating bedside cardiac ultrasound program in emergency medicine residency: a retrospective observation study from the perspective of performance improvement

PONE-D-20-30438R3

Dear Dr. Jung,

We’re pleased to inform you that your manuscript has been judged scientifically suitable for publication and will be formally accepted for publication once it meets all outstanding technical requirements.

Kind regards,

Robert Ehrman, MD, MS

Academic Editor

PLOS ONE
---

## [Editor Report · Acceptance letter]

24 Mar 2021

PONE-D-20-30438R3 

Operating bedside cardiac ultrasound program in emergency medicine residency: a retrospective observation study from the perspective of performance improvement 

Dear Dr. Jung:

I'm pleased to inform you that your manuscript has been deemed suitable for publication in PLOS ONE. Congratulations! Your manuscript is now with our production department. 

Kind regards, 

on behalf of

Dr. Robert Ehrman 

Academic Editor

PLOS ONE